# Enhancing low energy reconstruction and classification in KM3NeT/ORCA with transformers

**Iván Mozún Mateo[1*] on behalf of the KM3NeT collaboration**

**1** Laboratoire de Physique Corpusculaire de Caen

⋆ [mozun@lpccaen.in2p3.fr](mailto:mozun@lpccaen.in2p3.fr)

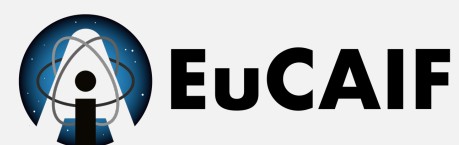

*The 2nd European AI for Fundamental Physics Conference (EuCAIFCon2025) Cagliari, Sardinia, 16-20 June 2025*

## Abstract

**The current KM3NeT/ORCA neutrino telescope, still under construction, has not yet reached its full potential in neutrino reconstruction capability. When training any deep learning model, no explicit information about the physics or the detector is provided, thus they remain unknown to the model. This study leverages the strengths of transformers by incorporating attention masks inspired by the physics and detector design, making the model understand both the telescope design and the neutrino physics measured on it. The study also shows the efficacy of transformers on retaining valuable information between detectors when doing fine-tuning from one configurations to another.**

## 1 Introduction

KM3NeT/ORCA is a neutrino telescope [1] placed in the Mediterranean sea, at a depth of 2450 km about 40 km from Toulon (France), with the objective of measuring the neutrino mass hierarchy using atmospheric neutrinos [2]. The detector design is a tridimensional array of PhotoMultiplier Tubes (PMTs) hosted within Digital Optical Modules (DOMs) [3] arranged along vertical detection units (DUs). The telescope collects light from Cherenkov photons emitted along the path of charged particles in neutrino interactions, creating a time-ordered sequence of light pulses with position and timing information that can be used to reconstruct neutrino event kinematics.

In this study, a novel deep learning architecture named transformer [4], that handles sequential data as those observed in a neutrino telescope is presented. In section 2, the model is introduced and the use of attention mask inspired by physics and detector design is motivated. In section 3, the challenges of reconstructing neutrino physics, both due to the physics itself and the limited telescope size, and how transformers overcome them are described.

## 2   A transformer model for neutrino telescopes

Thanks to the light pattern originated from a neutrino interaction (fig. 1), transformers are very well suited for reconstructing physics in neutrino telescopes.

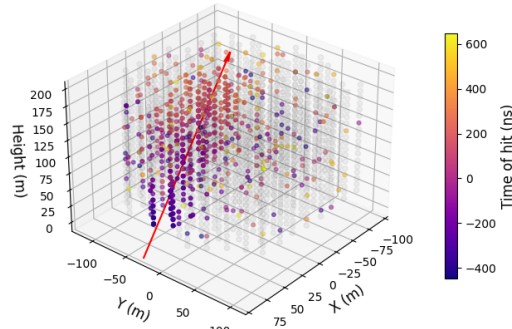

**Figure 1:** Example of a $\nu_\mu^{CC}$ event in KM3NeT/ORCA115: color dots represent the different light pulses in the PMTs and the heatmap is the arrival time w.r.t. to the mean time of arrival of all triggered hits in the event. Red solid line is the track $\mu$ produced after the neutrino interaction.

In KM3NeT/ORCA telescope, the raw observations can be arranged as a time-ordered sequence of light pulses with position and time information from the PMT that recorded it. A transformer model handles sequential data and processes their components in parallel, being able to capture complex patterns and establish relationships between the pulses through the use of the self-attention mechanism [4],

$$\texttt{Attention}(Q,K,V) = \texttt{SoftMax}\left(\frac{Q \cdot K^T}{\sqrt{d}}\right) \cdot V \tag{1}$$

where $Q$, $K$ and $V$ are matrices built from the input data, and $d$ is the latent space size of the model. If no prior information is given to the model, eq. (1) provides information about how the components of the input sequence are correlated among themselves. Nevertheless, no information about the physics is directly injected in the model.

To overcome this burden, physics constraints and detector information are artificially included in eq. (1) through the use of attention masks $U$ [5,6],

$$\texttt{PairwiseAttention}(Q,K,V;U) = \texttt{SoftMax}\left(\frac{Q \cdot K^T}{\sqrt{d}} + U\right) \cdot V \tag{2}$$

These attention masks are $N \times N$ matrices with $N$ being the input sequence length, that encode different correlations between the light pulses of an event, for instance, the space-time relativistic distance to identify pulses from the same source [6], the euclidean distance to quantify spatial proximity, or a local-coincidence mask to determine which pulses come from the same PMT, DOM or DU. This approach also allows into discrimination optical background hits from those coming from a physics source, called triggered hits, while increasing context length.

## 3   Challenges of neutrino physics reconstruction in KM3NeT/ORCA

When talking about reconstruction in a neutrino telescope, the main difficulty found is based on its detection principle. Since the PMTs detects light, only charged particles can easily be reconstructed, whereas non-charged particles are invisible to it, leading to an intrinsic bias when trying to reconstruct the whole interaction.

### 3.1 A growing telescope

The KM3NeT/ORCA telescope grows in size by adding DUs to already deployed ones, increasing its capability and sensitivity to measure neutrino properties. When a deep learning model is initialized, it does not contain physics information, it learns as it trains. However, this is not optimal because a model can already have learnt that information from a previous configuration. Besides, if a model is first trained on a larger telescope it retains valuable information from DUs not yet deployed as shown in fig. 2 by the area under the Receiver Operating Characteristic (ROC) curve, a metric that indicates the model's ability to distinguish the two neutrino events: $\nu_\mu^{CC}/\nu_e^{CC}$.

Figure 2 shows an improvement of over 20% achieved with a very limited training sample (100 events per class), by using interpolated information from a larger configuration, contrary to a model trained from scratch, whose performances are only comparable when using a large training sample (1M events per class), where limitations from the detector itself play a higher role rather than the model capabilities.

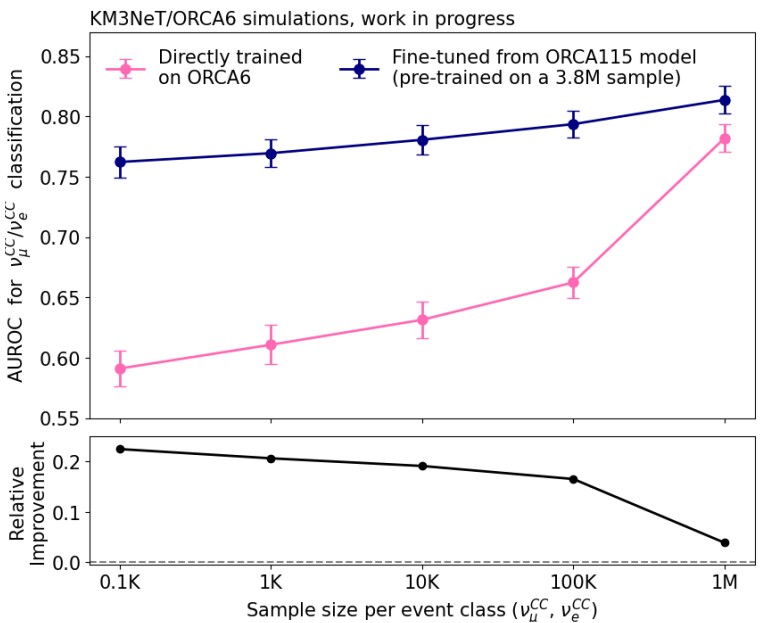

**Figure 2:** Evolution of the AUROC value for $\nu_\mu^{CC}/\nu_e^{CC}$ classification as a function of the training data size in ORCA6 for a model fine-tuned from ORCA115 (purple line) and a model trained from scratch (pink line). Modified from [7].

Another benefit of the use of deep learning for classification is that the training does not rely on any reconstructed variable, it computes the score it from what the raw data. On the contrary, classical classification methods rely on decision-tree-based classifiers trained using reconstruction variables. These techniques are prone to be strongly biased by reconstruction algorithms and must be updated whenever a new version of the reconstruction software is released.

### 3.2 Reconstructing neutrino energy and direction from light

The major difficulty of neutrino telescopes is that they are built to measure neutrino properties, but neutrinos are invisible to them. Therefore, reconstruction algorithms based on a maximum-likelihood fit (MLF) are developed to reconstruct only what is observed, which in most cases is based on either a track or a shower hypothesis, whereas in reality it is a mix of many things for which a likelihood is hard to define. Another limitation also arises from the

fact that non-visible particles are involved, leading to an intrinsic bias when reconstructing the neutrino energy from the outgoing particles.

A advantage of transformers compared to MLF algorithms is that they can be trained to fit any hypothesis, which means they can, in principle, reconstruct anything observed by the telescope. Also, if we train the transformer on all types of events[1], it can handle complex cases better. For instance, a high-energy $\nu_\mu^{CC}$ event that has both tracks and stochastic energy losses in the form of showers. Since the transformer has seen both tracks and showers during training, it can learn to reconstruct them more accurately whenever they appear in a new event and the inference time is significantly reduced since one single reconstruction is considered.

Therefore, as long as the neutrino properties are present in the training data, a deep learning model is trained to reconstruct them and to establish the necessary likelihood to extrapolate the neutrino properties from the observations in the telescope. Figure 3 demonstrates the improvement in direction (over 20%) and energy estimation w.r.t. a MLF algorithm when using transformers.

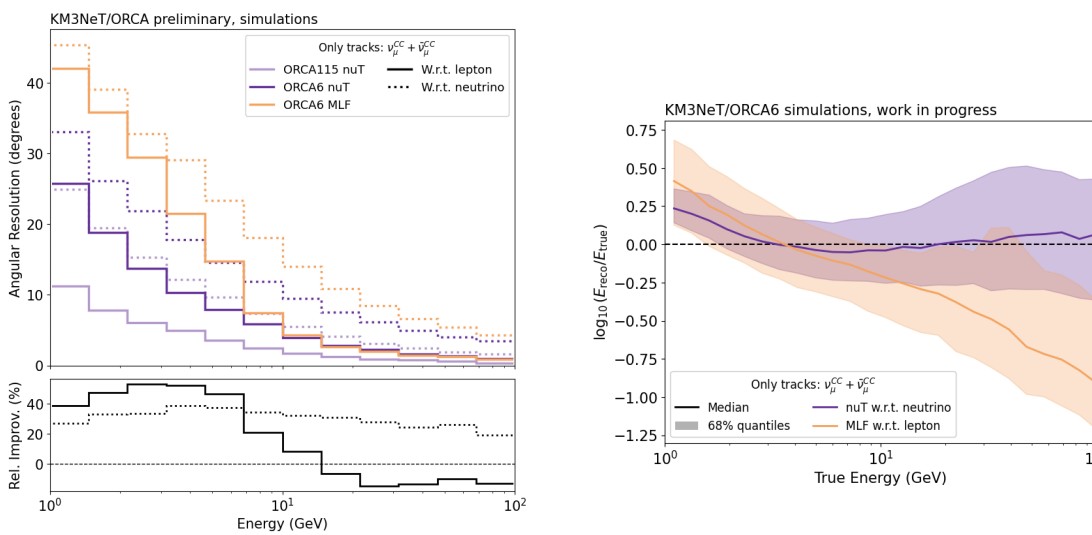

**Figure 3:** Left: angular resolution in direction reconstruction for ORCA6 (dark purple) and ORCA115 (light purple). Right: log 10 of ratio between reconstructed and true energy. Comparison to a MLF reconstruction algorithm is also shown in orange color.

# 4 Conclusion

Transformers are cutting edge deep learning models used for reconstruction of neutrino physics in neutrino telescopes. Through the use of, physics and detector-inspired attention masks, significantly improvements in neutrino reconstruction in simulations for the KM3NeT/ORCA telescope are shown. Compared to maximum-likelihood fits algorithms, limited by the defined likelihood, transformers achieve better direction and energy resolution of the neutrino at low energies, crucial for the study of neutrino oscillations. To leverage the limitations of training sample size, it is shown that the use of pre-trained models in larger configurations retains essential physics information and reduces the cost of training models, being a key for a detector that is under construction.

---

[1]For this work, only $CC$ interaction processes are considered.

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
