# Peer review of "Enhancing low energy reconstruction and classification in KM3NeT/ORCA with transformers"

_SciPost Physics Proceedings_

## Round 1 · Referee Report · Thomas Vuillaume (Referee 1) · 2025-12-5

Disclosure of Generative AI use

The referee discloses that the following generative AI tools have been used in the preparation of this report:

chatGPT 5.1, 05/12/2025 - spelling and prephrasing

Strengths

  1. Physics-informed attention masks. This is an interesting idea to inject physics into the transformer architecture.
  2. Transfer learning across configurations. This is highly relevant to KM3NeT but also to other experiments with different and/or growing configurations.

Weaknesses

  1. no study of the impact of the PairwiseAttention compared to standard Attention. Although the idea seems interesting, experimental proof of its validity and impact would be expected.

Report

The manuscript presents a transformer-based approach for the event reconstruction and classification in KM3NeT/ORCA. It is tested on simulated data and studies the performance improvement when trained on a larger telescope configuration and fine-tuned on a smaller one.
I recommended it for publication with minor changes.

Requested changes

  1. "PMTs detects light" p.2 -> "PMTs detect light"
  2. "it computes the score it from what the raw data", p.3 - rephrase
  3. "These techniques ... must be updated whenever a new version of the reconstruction software is released.", p.3 - I suppose the approach proposed here still requires a calibration step (not discussed in the paper) which also makes it dependent on the reconstruction software?
  4. "A advantage of transformers ", p.4 -> "An .."
  5. " the inference time is significantly reduced" - compared to what?

Recommendation

Ask for minor revision

---

## Editorial Decision

awaiting_resubmission